# A Needle In A Haystack: Referring Hour-Level Video Object Segmentation

## Abstract

Long-term videos over minutes are ubiquitous in daily life while existing Referring Video Object Segmentation (RVOS) datasets are limited to short-term videos with a duration of only 5-60 seconds. To unveil the dilemma of referring object segmentation towards hour-level videos, we construct the first Hour-level Referring Video Object Segmentation (Hour-RVOS) dataset characterized by (1) any-length videos from seconds to hours, (2) rich-semantic expressions with double length, and (3) multi-round interactions according to target change. These unique characteristics further bring tough challenges including (1) **Sparse object distribution**: Segmenting target objects in sparse-distributed key-frames from massive amounts of frames is like finding a needle in a haystack. (2) **Long-range correspondence**: Intricate linguistic-visual associations are required to establish across thousands of frames. To address these challenges, we propose a semi-online hierarchical-memory-association RVOS method for building cross-modal long-range correlations. Through interleaved propagation of hierarchical memory and dynamic balance of linguistic-visual tokens, our method can adequately associate multi-period representations of target objects in a real-time way. The benchmark results show that existing offline methods have to struggle with hour-level videos in multiple stages, whereas our proposed method without LLMs can achieve over 15% accuracy improvements compared to Sa2VA-8B when handling any-length videos with multi-round and various-semantic expressions in one-stage.

## 1 Introduction

The task of Referring Video Object Segmentation (RVOS) aims at segmenting the target objects specified by natural language expressions. Existing RVOS benchmarks (Seo et al., 2020; Ding et al., 2023; Liang et al., 2025a; Khoreva et al., 2019; Gavrilyuk et al., 2018) usually contain 5-60 seconds videos without switch of different scenes, and briefly object-describing expressions lacking of diverse semantics. Meanwhile, due to short durations, each target object can be segmented by pairing it with one expression as the initial reference, while the initial-expression cannot correspond to constantly changing targets in longer videos.

To this end, we construct a Hour-level Referring Video Object Segmentation (Hour-RVOS) dataset which contains 300 videos with 100.4h in total and 9114 expressions with 18.3 average words. The main three characteristics of our Hour-RVOS dataset are as follows: (1) **Any-Length Videos.** The duration of videos ranges from seconds to hours as shown in Fig. 1 (a), the average duration achieves 1204.8 seconds which is far longer than the one of any-existing RVOS datasets. Besides, these videos involve different scene/view switches which are not available in existing RVOS datasets. (2) **Rich-Semantic Expressions.** The rich-semantic expressions cover descriptions of appearance, motion and relationships as shown in Fig. 2. As the semantic complexity significantly increases, the average words of expressions achieve 18.3 which is more than twice the ones in existing expressions as illustrated in Fig. 1 (c). (3) **Multi-Round Interactions.** There are multiple expressions at multiple timestamps in each video to support multi-round human interactions as shown in Fig. 1 (e).

These unique characteristics of our Hour-RVOS dataset further bring tough challenges to RVOS field as follows: (1) **Sparse object distribution.** In our Hour-RVOS dataset, there are not only densely-distributed objects like the main character in the movie who appears in most frames, but also sparsely-distributed objects, for example, only appear in dozens of frames in videos with thou-

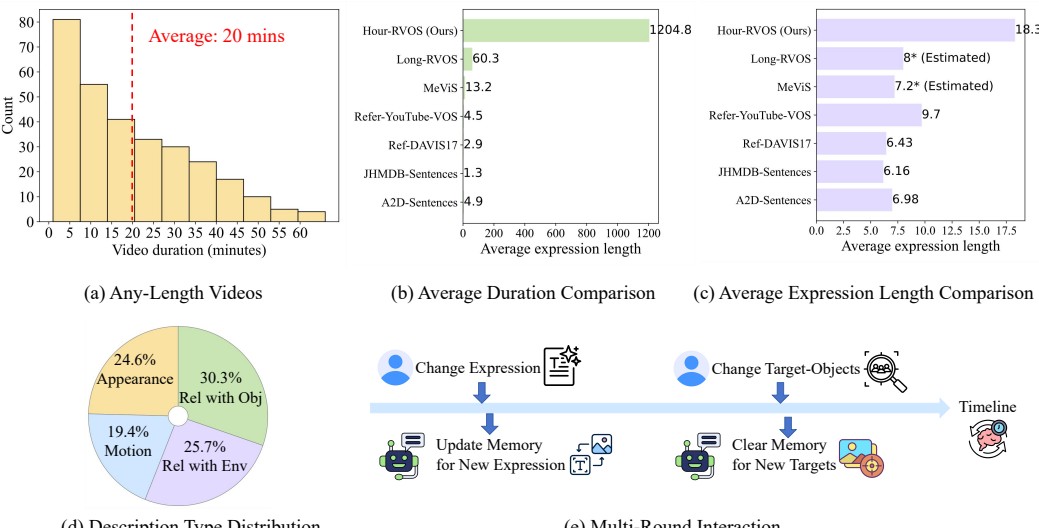

Figure 1: Characteristics of our Hour-RVOS dataset: (a) Any-Length Videos, (b) Average Duration Comparison, (c) Average Expression Length Comparison, (d) Description Type Distribution and (e) Multi-Round Interaction.

sands of frames. We calculate the average ratio of the duration that target objects appear in videos to the total video duration, and the result is as low as $13.7\%$ in the videos over 30 minutes. Therefore, for these target objects with sparse distribution in super-long videos, how to find the key frames from massive video frames becomes the most severe challenge. (2) **Long-range correspondence.** Hour-level videos consist of different clips containing essential and correlative object informations, thereby associating language descriptions and video clips in long-range is vital to discern target objects from other distractors. As shown in Fig. 2 (b), being picked up by a hand in the early stage is indispensable for distinguishing different cans in the later stage. Unlike the direct appearance/motion descriptions that correspond to each clip, the relationship descriptions can span multiple clips. Therefore, it is undesirable to simplify the super-long video segmentation into multi-clip segmentation. Simply processing different clips independently cannot establish long-range visual-linguistic correspondence to enhance segmentation performance.

To address these challenges, we propose a semi-online Hierarchical-Memory-Association RVOS method, termed as Memory-RVOS, to establish cross-modal long-range correlations. Specifically, we design a Hierarchical Memory Interleaved Propagation module to retain core informations of target objects when facing any-length videos. When handling videos in clip-by-clip mode (also called semi-online mode), the construction of hierarchical memory enables our method to effectively utilize the target information in the previous clips to enhance the segmentation effect of the current clip. Meanwhile, to deal with the semantic imbalance between linguistic and visual tokens which leads to the unsatisfactory association, we design a Linguistic-Visual Dynamic Balance module to update crucial corresponding tokens between multi-modals. Through associating target objects with language expressions in semi-online mode, our proposed Memory-RVOS method can segment the target objects in a real-time way. We conduct extensive experiments to benchmark existing RVOS methods (Yuan et al., 2024; He & Ding, 2024; Wu et al., 2023; Liang et al., 2025b) and Multimodal Large Language Models (MLLM) (Yan et al., 2024a; Gong et al., 2025; Lin et al., 2025; Yuan et al., 2025) that can handle RVOS task on our proposed Hour-RVOS dataset. Benchmark results sufficiently demonstrate the challenges are significantly difficult to be dealt with, whereas our proposed Memory-RVOS method can make a breakthrough to effectively address these challenges.

## 2 HOUR-RVOS

### 2.1 DATASET CONSTRUCTION

To ensure the complexity and richness of our Hour-RVOS dataset, we carefully select 300 videos with challenging objects from existing video object segmentation/tracking/video-language under-

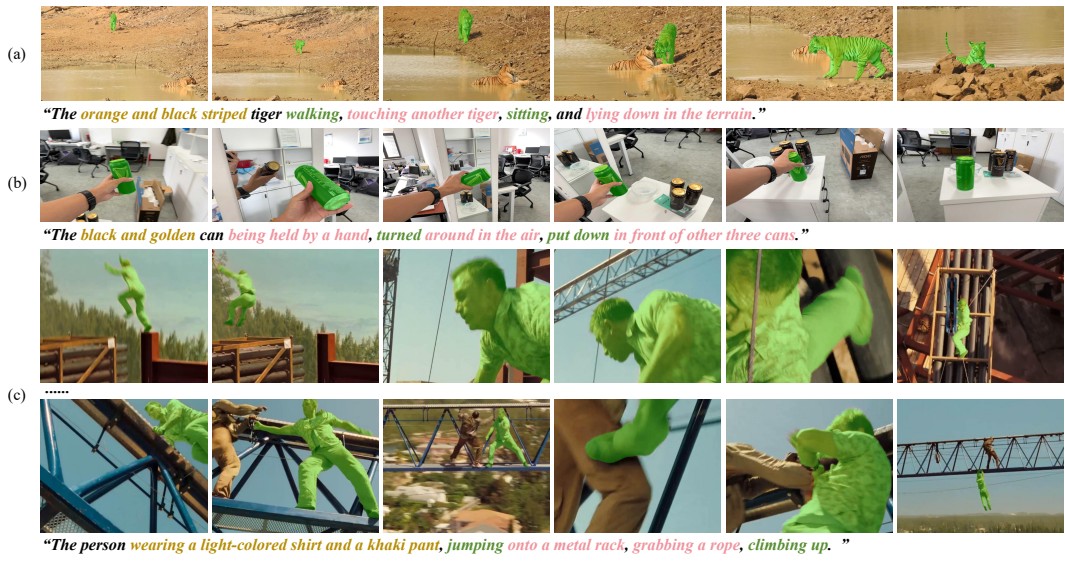

(a) *"The orange and black striped tiger walking, touching another tiger, sitting, and lying down in the terrain."*

(b) *"The black and golden can being held by a hand, turned around in the air, put down in front of other three cans."*

(c) ......
*"The person wearing a light-colored shirt and a khaki pant, jumping onto a metal rack, grabbing a rope, climbing up."*
......
*"The person with a white, patterned shirt, climbing on a metal scaffold, fighting with another person wearing brown clothes in the air."*

Figure 2: Examples from our proposed Hour-RVOS dataset including (a-b) clips from videos with their expressions respectively, (c) multiple clips from the same video with multi-round expressions in sequence. The brown, green and pink words in expressions denote the descriptions in terms of appearance, motion and relationship respectively.

standing datasets (Hong et al., 2023; 2024; Kristan et al., 2023; Yang et al., 2022a;b; Tang et al., 2023a; Chandrasegaran et al., 2024; Hu et al., 2023). The selected videos are from different domains (e.g., records of daily lives, sports, cookings, movies, cartoons, documentaries) and viewpoints (ego. and exo.). The objects in these videos have significant changes at the aspects of appearance, motion and relationship with environment/objects.

After identifying these videos and the target objects, we equip these target objects with high-quality masks. Specifically, we first generate the bounding box of target objects with GoundingDINO (Liu et al., 2024) in each frame that objects appear, then utilize SAM2 (Ravi et al., 2024) to generate initial masks based on the bounding boxes, finally hire annotators to manually correct these initial masks to obtain the high-quality masks.

To generate complex semantic expressions for these target objects, we firstly divide the entire long videos into clips based on scene with PySceneDetect (Castellano, 2025), then we generate object descriptions in different aspects (appearance, motion and relationship) independently with Multi-Modal Large Language Models (MLLMs) (Hurst et al., 2024; Zhu et al., 2025). Moreover, we adopt LLM (Hurst et al., 2024) to merge descriptions of target objects in different aspects to generate the expressions with complex semantics. Through manual corrections and final confirmations, we generate high-quality expressions of target objects in these long videos, and construct the first Hour-level RVOS dataset containing 9114 high-quality expressions with rich semantics for target objects in 300 videos with duration from seconds to hours.

## 2.2 DATASET ANALYSIS AND STATISTICS

We comprehensively analyze the characteristics of our proposed Hour-RVOS dataset by comparing with the existing representative Referring Video Object Segmentation datasets (Gavrilyuk et al., 2018; Khoreva et al., 2019; Seo et al., 2020; Ding et al., 2023; Liang et al., 2025a) in Tab. 1.

**Video Statistics.** As shown in Fig. 1 (a), there are 300 videos which duration range from second-level to hour-level. The average duration of these videos achieves 1204.8s, which is much longer than the average duration of existing RVOS datasets, for example, the mean duration is only 60.3s in Long-RVOS dataset (Liang et al., 2025a). As shown in Tab. 1, in terms of total duration, mean duration, mean frame, max frame and masks, our proposed Hour-RVOS dataset far exceeds the ones in existing RVOS datasets. Moreover, we calculate the ratio of the duration target objects appear in videos to the total duration as shown in Fig. 3 (a). As the duration of videos increases,

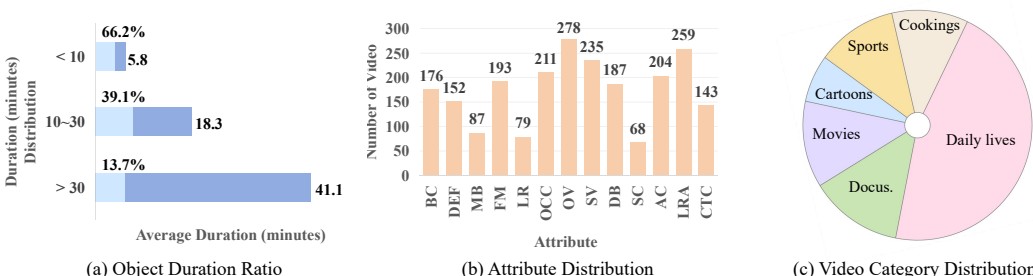

Figure 3: Statistics of our proposed Hour-RVOS dataset. (a) The ratio of objects appearing in videos of different lengths, (b) the distribution of target-object attributes (the definition of these attributes are listed in Appendix A.4.), (c) the distribution of video categories.

Table 1: Statistical comparison between our proposed Hour-RVOS and existing RVOS datasets. ∗ denotes estimating statistics on the publicly datasets, and Expn. is the abbreviation of expression.

| Dataset | Videos | Expn. | Total Duration | Mean Duration | Mean Frame | Max Frame | Masks | Objects | Object Classes | Avg. Words | Multiple Round |
|---|---|---|---|---|---|---|---|---|---|---|---|
| A2D-Sentence (Gavrilyuk et al., 2018) | 3782 | 6656 | 4.2h | 4.9s | 3.2 | 84 | 58k | 4825 | 6 | 6.98 | ✗ |
| JHMDB-Sentence (Gavrilyuk et al., 2018) | 928 | 928 | 0.3h | 1.3s | 34 | 40 | 32k | 928 | 1 | 6.16 | ✗ |
| Ref-DAVIS$_{17}$ (Khoreva et al., 2019) | 90 | 1544 | 0.1h | 2.9s | 69 | 104 | 14k | 205 | 78 | 6.43 | ✗ |
| Ref-YouTube-VOS (Seo et al., 2020) | 3978 | 15009 | 5.0h | 4.5s | 27 | 36 | 131k | 7451 | 94 | 9.7 | ✗ |
| MeViS (Ding et al., 2023) | 2006 | 28570 | 7.3h | 13.2s | 79 | - | 443k | 8171 | 36 | 7.2∗ | ✗ |
| Long-RVOS (Liang et al., 2025a) | 2193 | 24689 | 36.7h | 60.3s | 362 | - | 2.1M | 6703 | 163 | 8∗ | ✗ |
| Hour-RVOS (Ours) | 300 | 9114 | **100.4h** | **1204.8s** | **7229** | **23937** | **2.4M** | 3873 | 97 | **18.3** | ✔ |

the ratio gradually decreases to only 13.7% in the videos over 30 minutes. We count the attribute distributions defined in (Hong et al., 2024) (The complete definitions are listed in Appendix A.4) as shown in Fig. 3 (b). Among these attributes, our videos contain the most Out-of-View (OV) and Long-term Reappearance (LRA) attributes with exceeding 250 videos corresponding to them. The comprehensive distribution of attributes indicate the complexity and challenges in our Hour-RVOS dataset. We further count the proportion of video categories as shown in Fig. 3 (c), our 300 videos cover records of daily lives, movies, cartoons, sports, cookings and documentaries.

**Expression Statistics.** The 9114 expressions in our Hour-RVOS dataset contain rich semantics including the descriptions of appearance, motion, relationships with environments/other objects. Therefore, the average words of our expressions achieve 18.3 which is more than twice the number of average words of expressions in the existing RVOS datasets as shown in Tab. 1. Meanwhile, our Hour-RVOS dataset is no longer limited to one video corresponding to one initial expression. As the scene in videos changes, the expression assigned to the target objects is also adjusted, thus achieving accurate linguistic-visual correspondence. As our dataset requires RVOS methods to consider the impact of multiple expressions in one video, the RVOS method trained with our dataset can support multi-round human interactions where users can change the expression at any time.

## 3 MEMORY-RVOS

### 3.1 TASK DEFINITION

Given a video $V = \{I_t\}_{t=1}^{T}$ as a sequence of $T$ frames, a sequence of $N_p$ interactive prompts $P = \{(E_j, \tau_j)\}_{j=1}^{N_p}$ where $E_j$ denotes the $j$-th natural language expression referring to $N$ target objects, $\tau_j \in \{1, ..., T\}$ denotes the timestamp at which expression $E_j$ is provided. The goal of RVOS method $F$ is continuously output a set of segmentation masks $M = \{\{m_{k,t}\}_{k=1}^{N}\}_{t=1}^{T}$ denoting the sequence of binary masks for each target object across all frames which can be formulated as:

$$F(V, P) \rightarrow \{M_k\}_{k \in N}. \tag{1}$$

### 3.2 TOWARDS ANY-LENGTH VIDEOS WITH MULTI-ROUND EXPRESSIONS

In semi-online mode, we first divide the video with $T$ frames into $K$ clips evenly, where the size of each clip is $\lfloor T/K \rfloor$. As shown in Fig. 4 (a), each input consists of a clip $C_i$ and an expression $E_j$ with

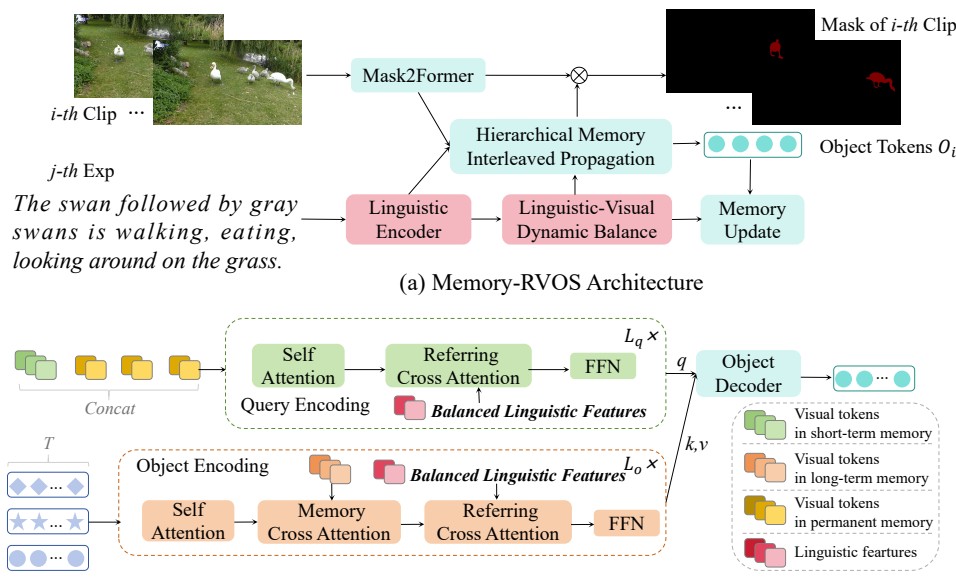

Figure 4: (a) Overview of our proposed semi-online Memory-RVOS method. (b) our Hierarchical Memory Interleaved Propagation module.

$L_j$ words. After encoding the expression with the linguistic encoder, we integrate linguistic features $Z_j \in \mathbb{R}^{L_j \times d}$ and object queries $Q \in \mathbb{R}^{N_q \times d}$ with cross-attention as $\hat{Q} = Q + \text{softmax}(\frac{QZ^T}{\sqrt{d}})Z$, where $N_q$ is the number of queries. We send integrated queries $\hat{Q}$ and frames in clip $C_i$ into the Mask2Former (Cheng et al., 2022) to extract potential object tokens and frame-level mask features. Then we associate visual-linguistic tokens in our proposed Hierarchical Memory Interleaved Propagation module. Finally, we use these target-object tokens to produce object mask by aggregating with frame-level mask features, and update our permanent-long-short hierarchical memory.

When a new expression arrives, we input the encoded features $Z_j, Z_{j+1}$ of the new and previous expressions $E_j, E_{j+1}$ into our proposed Linguistic-Visual Dynamic Balance module. Through calculating with visual tokens in hierarchical memory, we adjust the attention weights of linguistic features and remove outdated visual tokens. With the construction of hierarchical memory, the queries for the current clip can adequately integrate historical informations of target objects, which is essential for associating the current expression with target objects over a long-time span.

### 3.3 HIERARCHICAL MEMORY INTERLEAVED PROPAGATION

**Design Principles.** In our memory design, short-term memory ensures object queries can be associated with the latest state of target objects, long-term memory ensures object queries can correlate complex descriptions and visual informations over a wider range of time scales, permanent memory ensures that when target objects disappear for a long time, so long that long-term memory does not contain valid target-object tokens, the stored target-object tokens in permanent memory are still available to match the target objects when they reappear in the video.

**Hierarchical Memory Update.** As shown in Fig. 5 (a), for the current input clip $C_i$, we update short-term memory with the generated target-object tokens $O_{i-1}$ when handling $C_{i-1}$ to memorize the latest target-related features. Meanwhile, we update long-term memory with the target-object tokens of past $W_l$ clips including $\{O_{i-2}, ..., O_{i-1-W_l}\}$ where $W_l$ denotes the window size of long-term memory. To extract permanent memory, we first calculate the similarity between the current linguistic features and stored tokens in long-term memory, then select the tokens $O_{i-k}$ which are most relevant and store them into the permanent memory where window size is set to $W_p$.

**Query Encoding.** As target objects could disappear for a while in long-term videos, the tokens in-term memory may not correspond to objects that re-appear in the current input clip, thus we first concatenate short-term tokens and compressed permanent tokens to ensure valid tokens as

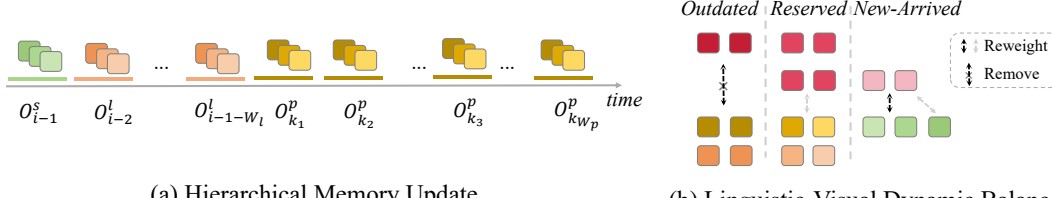

(a) Hierarchical Memory Update    (b) Linguistic-Visual Dynamic Balance

Figure 5: Illustrations of (a) Hierarchical Memory Update and (b) Linguistic-Visual Dynamic Balance processes in our Memory-RVOS.

shown in Fig. 4 (b). We compress permanent tokens with bipartite soft strategy method (Bolya et al., 2022). After splitting $n$ tokens into two non-overlap sets $\mathbb{A}$ with $r$ tokens and $\mathbb{B}$ with $n-r$ tokens, we calculate cosine similarity between tokens in two sets and select Top-$r$ token pairs with the highest similarity to merge, producing $n-r$ compressed permanent tokens. Then we send the concatenated tokens into $L_q$ cascaded blocks where each block consists of self-attention, referring cross-attention and FFN layers. Balanced linguistic features act as key-value in referring cross-attention.

**Object Encoding and Decoding.** After matching the current object tokens with Hungarian matching algorithm (Kuhn, 1955), we send generated object trajectories into $L_o$ cascaded blocks where each block consists of self-attention, memory cross-attention, referring cross-attention and FFN layers. The tokens in long-term memory act as key-value in memory cross-attention and the balanced linguistic features act as key-value in referring cross-attention. Finally, we send the outputs of object and query encoding process into object decoder to act as key-value and query respectively. The object decoder consists of regular self-attention, cross-attention and FFN layers and generate the final target-object tokens to filter out target-object masks.

Compared to propagating hierarchical memory in temporal sequence, in our proposed hierarchical memory interleaved propagation module, we place different memory tokens into different encoding process to adequately match visual-linguistic tokens in different temporal scale.

### 3.4 LINGUISTIC-VISUAL DYNAMIC BALANCE

**Imbalance Issue.** In memory-based method, the number of visual tokens stored in memory far exceeds the number of linguistic tokens. Among visual tokens, there are tokens irrelevant to the current expression which act as noise to affect current queries. To balance the quantity of tokens, we design a Linguistic-Visual Dynamic Balance module to prune as much noise visual tokens as possible from memory and reweight linguistic features to highlight valuable visual tokens.

**Expression Correlation Partition.** When a new expression $E_{j+1}$ arrives at $\tau_{j+1}$ timestamp, we firstly calculate the cosine similarity on the encoded linguistic features $Z_j, Z_{j+1}$ between the previous expression $E_j$ and the new expression $E_{j+1}$. Based on the similarity score, we divide the linguistic features $Z_j, Z_{j+1}$ into three types including outdated, reserved, new-arrived as shown in Fig. 5. We regard the similar parts of $Z_j, Z_{j+1}$ as reserved parts by setting a hyper-parameter similarity threshold. Excluding the reserved parts, the remaining parts in $Z_j$ are considered as outdated parts, and the remaining parts in $Z_{j+1}$ are considered as new-arrived parts. After clarifying different parts, we use the outdated linguistic features to prune the noise permanent tokens which are irrelevant with the new expression. Meanwhile, we use the new-arrived linguistic features to highlight the visual tokens store in long-term and short-term memory which are effective to capture the changed target objects based on the new-arrived descriptions.

**Prune Noise-tokens.** Supposed there are $n$ tokens stored in permanent memory, we prune top-$k$ noise tokens based on the similarity between permanent tokens and the outdated linguistic features. When the storage exceeds the window size of permanent memory before new expression arrives, stored tokens are eliminated according to First-In-First-Out strategy.

**Reweight Linguistic-tokens.** We reweight the linguistic features of the new expression by firstly calculating the cosine similarity denoted as $s$ between original and new linguistic features. Then we multiply $1 - s$ with the new linguistic features as $\hat{Z}_{j+1} = (1 - s) \cdot Z_{j+1}$. Finally, we send the balanced the linguistic features $\hat{Z}_{j+1}$ into the query and object encoding process.

Table 2: Benchmark of existing SOTA RVOS methods and our method on our proposed Hour-RVOS test set including three subsets divided by duration.

| Method | Type | Backbone | FPS | Test (0-10 mins) | | | Test (10-30 mins) | | | Test (over 30 mins) | | |
|---|---|---|---|---|---|---|---|---|---|---|---|---|
| | | | | $\mathcal{J}\&\mathcal{F}$ | $\mathcal{J}$ | $\mathcal{F}$ | $\mathcal{J}\&\mathcal{F}$ | $\mathcal{J}$ | $\mathcal{F}$ | $\mathcal{J}\&\mathcal{F}$ | $\mathcal{J}$ | $\mathcal{F}$ |
| Multi-Stage (Referring Segmentation + Video Propagation with SAM2 (Ravi et al., 2024)) | | | | | | | | | | | | |
| SOC (Luo et al., 2023) | Offline | Video-Swin-B | - | 14.6 | 9.7 | 19.5 | 11.6 | 9.1 | 14.1 | 9.4 | 5.8 | 13.0 |
| LMPM (Ding et al., 2023) | Offline | Video-Swin-B | - | 14.2 | 9.3 | 19.1 | 11.4 | 8.5 | 14.3 | 9.3 | 6.1 | 12.5 |
| DsHMP (He & Ding, 2024) | Offline | Video-Swin-B | - | 16.0 | 11.2 | 20.8 | 13.1 | 9.4 | 16.8 | 11.5 | 8.6 | 14.4 |
| MUTR (Yan et al., 2024b) | Offline | Video-Swin-B | - | 18.9 | 13.5 | 24.3 | 15.0 | 11.3 | 18.7 | 12.9 | 10.2 | 16.6 |
| ReferDINO (Liang et al., 2025b) | Offline | Video-Swin-B | - | 19.8 | 14.1 | 25.5 | 16.6 | 12.7 | 20.5 | 13.4 | 10.8 | 16.0 |
| One-Stage | | | | | | | | | | | | |
| VISA (Yan et al., 2024a) | Offline | Chat-UniVi-7B | 9 | 22.8 | 21.2 | 24.4 | 17.4 | 13.0 | 21.8 | 15.1 | 11.7 | 18.5 |
| VRS-HQ (Gong et al., 2025) | Offline | Chat-UniVi-7B | 5 | 24.6 | 22.4 | 26.8 | 19.2 | 14.7 | 23.7 | 15.5 | 13.8 | 17.2 |
| GLUS (Lin et al., 2025) | Offline | LLaVA-7B | 7 | 23.4 | 21.3 | 25.5 | 18.9 | 14.3 | 23.5 | 15.9 | 14.0 | 17.8 |
| Sa2VA (Yuan et al., 2025) | Offline | InternVL2.5-8B | 8 | 25.8 | 22.2 | 29.4 | 21.4 | 17.5 | 25.3 | 16.3 | 14.1 | 18.5 |
| OnlineRefer (Wu et al., 2023) | Semi-Online | Video-Swin-B | 26 | 20.3 | 16.1 | 24.5 | 18.9 | 15.1 | 22.7 | 15.6 | 11.3 | 19.9 |
| Memory-RVOS (Ours) | Semi-Online | Video-Swin-B | 31 | **42.6** | **37.9** | **47.3** | **34.3** | **30.1** | **38.5** | **27.5** | **24.9** | **30.1** |

# 4 EXPERIMENTS

## 4.1 EXPERIMENT SETTINGS

We divide our proposed Hour-RVOS dataset containing 300 videos into train, validation and test sets which contain 180, 30 and 90 videos respectively. To ensure the fair distribution of each set, we make sure the class distributions of target objects in each set are consistent, and the duration of videos in each set are ranging from seconds to hours. We adopt the evaluation metrics including region similarity $\mathcal{J}$, contour accuracy $\mathcal{F}$ and their average $\mathcal{J}\&\mathcal{F}$ following the same evaluation settings (Khoreva et al., 2019; Seo et al., 2020; Ding et al., 2023). To clearly illustrate the impact of videos with different durations on segmentation results, we divide the test sets into 3 subsets based on the duration including 0-10, 10-30, over 30 minutes, each subset contains 30 videos.

**Existing offline methods (Liang et al., 2025b; Ding et al., 2023; He & Ding, 2024; Luo et al., 2023; Yan et al., 2024b) and MLLMs (Yan et al., 2024a; Lin et al., 2025; Gong et al., 2025; Yuan et al., 2025) cannot directly handle long-term videos due to their huge GPU memory usage, and address multiple-expressions for one video limited by the setting that one expression corresponds to an entire video.** Therefore, we first divide the videos into clips based on different timestamps at which expressions arrive. For clips that can be dealt with at one time, we evaluate these methods according to original settings. For clips with so many frames that offline methods cannot process at once, we perform evaluation in two settings: (1) **Multi-Stage.** For these offline RVOS methods (Yan et al., 2024b; Ding et al., 2023; Luo et al., 2023; He & Ding, 2024; Liang et al., 2025b), we first uniform-sample frames from these clips for referring segmentation, then we propagate masks of sampled frames to the entire clip. (2) **One-Stage.** For these MLLMs (Yan et al., 2024a; Gong et al., 2025; Lin et al., 2025; Yuan et al., 2025) which contains mask propagation process, we follow their original keyframe selection strategies (We adjust the initial five-frame selection strategy of Sa2VA to uniform-sample strategy). We keep the same number of selected keyframes and uniform-sampled frames for fair comparison. **In contrast, our Memory-RVOS can process any-length videos and multi-round expressions under one stage in real-time way.**

**Evaluation results of our proposed Memory-RVOS method on existing RVOS benchmark including Ref-DAVIS17 (Khoreva et al., 2019), Ref-YouTube-VOS (Seo et al., 2020) and MeViS (Ding et al., 2023), training and inference details are shown in Appendix A.2 and A.3.**

## 4.2 HOUR-RVOS BENCHMARK RESULTS

As shown in Tab. 2, existing RVOS and MLLM methods (Wu et al., 2023; Yan et al., 2024b; Luo et al., 2023; He & Ding, 2024; Liang et al., 2025b; Yan et al., 2024a; Gong et al., 2025; Lin et al., 2025; Yuan et al., 2025) achieve unsatisfactory performance ranging from 9.3% to 25.8% in terms of $\mathcal{J}\&\mathcal{F}$ when evaluated on our proposed Hour-RVOS dataset, whereas these methods can achieve over 65% accuracy on Ref-YouTube-VOS (Seo et al., 2020). When testing the methods' performance on videos of different durations in a more fine-grained temporal dimension, the performance trend is that as the video duration increases, the segmentation accuracy significantly decreases.

For these RVOS methods (Wu et al., 2023; Luo et al., 2023; Yan et al., 2024b; He & Ding, 2024; Liang et al., 2025b), when the temporal span corresponding to the expressions increases significantly as the video length increases, these methods cannot deal with the object segmentation of long videos due to the lack of effective linguistic-visual correspondence across spatial and temporal dimension. Especially for these offline RVOS methods, they only consider the case of inputting all frames of the video at the same time when designing. During mutli-stage inference, the segmentation of target objects in most frames depends on the mask propagation with SAM2 (Ravi et al., 2024) which cannot establish the association across clips.

For these MLLM methods (Yan et al., 2024a; Gong et al., 2025; Lin et al., 2025; Yuan et al., 2025), although their ability of video understanding on short videos is competitive, as video length increases, the coverage of sampled frames decreases, which leads to missing the key information during video understanding, resulting in significant accuracy drops. The benchmark results adequately demonstrate the significant challenges posed by our Hour-RVOS dataset.

Compared to these RVOS and MLLM methods, our proposed semi-online Memory-RVOS method achieves the best performance. Hierarchical memory can reduce the forgetting of target-object informations, and linguistic-visual dynamic balance enables our method to adjust the focus of segmentation according to the new-arrived expression, thereby improving the performance of referring object segmentation when handling any-length videos. We further visualize failure cases in Appendix A.4 to intuitively show complexity of challenges.

Under the same condition that the number of input frames fills the GPU memory, our Memory-RVOS method can achieve real-time segmentation speed at 31 FPS benefit from the effective compression and filtering of target-object tokens stored in hierarchical memory. In contrast, these offline RVOS methods and MLLMs with large-scale parameters achieve unsatisfactory efficiency.

### 4.3 ABLATION STUDIES

We conduct ablation studies on the validation set which is divided into two subsets based on the duration including 0-10, over 10 minutes, containing 15, 15 videos respectively. More ablation studies on settings of clip size and window size are shown in Appendix A.5.

**Hierarchical Memory Interleaved Propagation.** Under semi-online mode, at least the object tokens of the previous clip are needed as the query for the next clip (Heo et al., 2023), thus we keep short-term memory in ablation studies as shown in Tab. 3. The accuracy continues to improve with the introduction of long-term and permanent memory. Permanent memory achieves larger improvements on long-term videos.

Table 3: Ablation studies on the hierarchical memory. SM, LM and PM denote Short-term, Long-term and Permanent Memory respectively.

| SM | LM | PM | Val (0-10 mins) | | | Val (over 10 mins) | | |
|---|---|---|---|---|---|---|---|---|
| | | | $\mathcal{J}\&\mathcal{F}$ | $\mathcal{J}$ | $\mathcal{F}$ | $\mathcal{J}\&\mathcal{F}$ | $\mathcal{J}$ | $\mathcal{F}$ |
| ✓ | | | 36.3 | 33.1 | 39.5 | 20.8 | 16.4 | 25.2 |
| ✓ | ✓ | | 39.8 | 35.5 | 44.1 | 24.2 | 20.7 | 27.7 |
| ✓ | | ✓ | 42.1 | 37.6 | 46.6 | 28.4 | 25.0 | 31.8 |
| ✓ | ✓ | ✓ | 45.8 | 40.9 | 50.7 | 31.2 | 27.5 | 34.9 |

**Linguistic-Visual Dynamic Balance.** We test the performance of our method under four settings: lack of balance, only pruning noise visual tokens based on outdated linguistic features (prune-only), only reweighting the attention of linguistic features (reweight-only) and the full balance. As shown in Tab. 4, both pruning noise and reweighting linguistic features are effective for performance improvements.

Table 4: Ablation studies on the linguistic-visual dynamic balance.

| Setting | Val (0-10 mins) | | | Val (over 10 mins) | | |
|---|---|---|---|---|---|---|
| | $\mathcal{J}\&\mathcal{F}$ | $\mathcal{J}$ | $\mathcal{F}$ | $\mathcal{J}\&\mathcal{F}$ | $\mathcal{J}$ | $\mathcal{F}$ |
| w/o balance | 41.6 | 36.1 | 47.1 | 25.3 | 21.7 | 28.9 |
| prune-only | 44.7 | 39.8 | 49.6 | 27.1 | 22.9 | 31.3 |
| reweight-only | 44.4 | 39.3 | 49.5 | 28.3 | 24.1 | 32.5 |
| w/ balance | 45.8 | 40.9 | 50.7 | 31.2 | 27.5 | 34.9 |

## 5 DISCUSSIONS

**How irrelevant frames affect the segmentation of long videos?** To further explore the affects of these irrelevant frames, we remove irrelevant clips of videos over 10 minutes in validation set (all frames in these clips do not contain target objects) at a ratio of $50\%$ and $90\%$, then test methods on these clip-removed videos and the results are shown in Tab. 5. As the proportion of removed-clips gradually increases, the segmentation accuracy of these method (Liang et al., 2025b; Yuan et al.,

Table 5: The impact of irrelevant clips and simplified expressions in long videos.

| Method | Val (over 10 mins) | | | Val (50%-remove) | | | Val (90%-remove) | | | Val (simplified) | | |
|---|---|---|---|---|---|---|---|---|---|---|---|---|
| | $\mathcal{J}\&\mathcal{F}$ | $\mathcal{J}$ | $\mathcal{F}$ | $\mathcal{J}\&\mathcal{F}$ | $\mathcal{J}$ | $\mathcal{F}$ | $\mathcal{J}\&\mathcal{F}$ | $\mathcal{J}$ | $\mathcal{F}$ | $\mathcal{J}\&\mathcal{F}$ | $\mathcal{J}$ | $\mathcal{F}$ |
| ReferDINO (Liang et al., 2025b) + SAM2 (Ravi et al., 2024) | 19.9 | 15.1 | 24.7 | 21.3 | 17.2 | 25.4 | 24.3 | 20.5 | 28.1 | 16.3 | 12.1 | 20.5 |
| Sa2VA (Yuan et al., 2025) | 28.4 | 23.6 | 33.2 | 30.6 | 26.3 | 34.9 | 35.7 | 33.1 | 38.3 | 26.4 | 21.9 | 30.9 |
| Ours | 31.2 | 27.5 | 34.9 | 33.7 | 30.1 | 37.3 | 36.1 | 33.6 | 38.6 | 30.6 | 27.7 | 33.5 |

2025) are also gradually increasing, especially for Sa2VA (Yuan et al., 2025) with stronger video understanding capabilities, the accuracy improvement is the most significant. Therefore, capturing target-object informations and filtering interference informations with hierarchical memory become essential, which is the key reason of hierarchical memory design in our Memory-RVOS method.

**How does our method perform when faced with simpler descriptions?** In realistic interactions, users may only provide simpler descriptions of target objects, thus we stimulate this situation to test the robustness of these methods. We adopt LLM (Hurst et al., 2024) to simplify the expressions in validation set from appearance/motion/relationship perspectives under the premise of expression correctness, and use the simplified expressions to test RVOS and MLLM methods (Liang et al., 2025b; Yuan et al., 2025). As shown in Tab. 5, the accuracy of all methods decreases as these methods are prone to segment similar distractors due to less information in simplified expressions, whereas our Memory-RVOS method has minimal drop in accuracy compared to other methods. Therefore, our proposed Memory-RVOS performs more robustly when facing simpler expressions.

# 6 RELATED WORKS

## 6.1 REFERRING VIDEO OBJECT SEGMENTATION DATASETS

In the past few years, A2D-Sentences and JHMDB-Sentences datasets (Gavrilyuk et al., 2018) are proposed to segment actors and their actions based on natural language sentences. Moreover, Ref-DAVIS17 (Khoreva et al., 2019) and Ref-YouTube-VOS (Seo et al., 2020) datasets are constructed based on existing video object segmentation datasets. Recently, MeViS (Ding et al., 2023), focusing more on the motion of objects, is constructed to highlight the object motion informations. Meanwhile, Long-RVOS (Liang et al., 2025a) is proposed, featuring long-term videos in which the average duration achieves 60 seconds. Compared to existing RVOS datasets, we explore the performance when facing any-length videos, rich-semantic expressions and multi-round interactions.

## 6.2 REFERRING VIDEO OBJECT SEGMENTATION METHODS

Representative RVOS methods including MTTR (Botach et al., 2022), ReferFormer (Wu et al., 2022), HTML (Han et al., 2023), TempCD (Tang et al., 2023b), SOC (Luo et al., 2023), SgMg (Miao et al., 2023), R2VOS (Li et al., 2023), LoSh (Yuan et al., 2024), SSA (Pan et al., 2025) and Refer-DINO (Liang et al., 2025b) introduce unique designs respectively into the transformer architecture for better object understanding and segmentation. Moreover, LMPM (Ding et al., 2023), DsHmp (He & Ding, 2024) and DMVS (Fang et al., 2025) focuses on object motions to promote performance. OnlineRefer (Wu et al., 2023), as a semi-online method, segments each frame or clip with cross-frame query propagation. In this work, we propose a semi-online hierarchical-memory-association method, which can build long-term associations in different clips of long-term videos.

# 7 CONCLUSIONS

In this work, we propose the first Hour-level Referring Video Object Segmentation dataset, and an innovative semi-online Hierarchical-Memory-Association RVOS method to address tough and unique challenges. We conduct comprehensive experiments to benchmark existing RVOS and MLLM methods on our Hour-RVOS datasets, and our Memory-RVOS method achieves significant performance improvement in a real-time way. The construction of Hour-RVOS dataset and Memory-RVOS method brings more inspiration and exploration space for the development of RVOS field.

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

# A APPENDIX

## A.1 THE USE OF LLMS

We use Large Language Models (LLMs) to aid or polish writing.

## A.2 TRAINING AND INFERENCE DETAILS

The main architecture of our proposed Memory-RVOS method adopts ViTA (Heo et al., 2022) following LMPM (Ding et al., 2023), DsHMP (He & Ding, 2024) and DMVS (Fang et al., 2025). We use Video-Swin (Liu et al., 2022) to encode the input $i$-th clip, and RoBERTa (Liu, 2019) as linguistic encoder to encode the input language expression. Following ViTA (Heo et al., 2022), and GenVIS (Heo et al., 2023), we use the frame-level loss $L_f$ to compute between the final frame-level

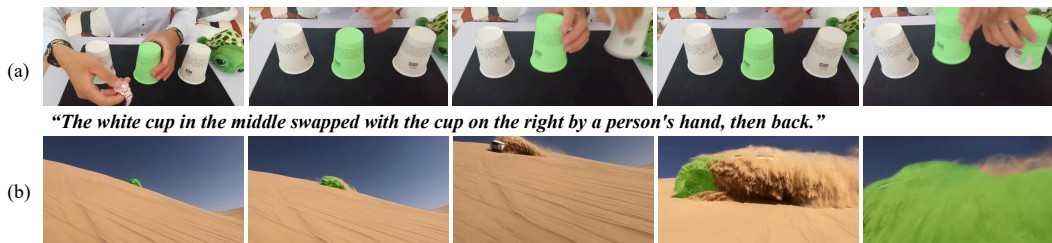

(a)

*"The white cup in the middle swapped with the cup on the right by a person's hand, then back."*

(b)

*"The vehicle being driven on the sand, kicking up sand, buried in the sand."*

Figure A1: Failure cases when evaluating our method on our proposed Hour-RVOS datasets.

output and the groud-truth mask, the video-level loss $L_v$ and the matching loss $L_m$ to associate the object across clips, thus the total loss $L = L_f + \lambda L_v + \mu L_m$. The hyperparameters $\lambda, \mu$ in training loss are set to 1 and 1 during training, the hyperparameter similarity threshold in the process of expression correlation partition is set to 0.8.

During benchmark, we re-train representative RVOS methods including OnlineRefer (Wu et al., 2023), LoSh (Yuan et al., 2024), DsHMP (He & Ding, 2024), ReferDINO (Liang et al., 2025b) only on the combining the train sets of our Hour-RVOS and MeViS (Ding et al., 2023) datasets at a ratio of 1 : 3. For these MLLMs including VISA (Yan et al., 2024a), VRS-HQ (Gong et al., 2025), GLUS (Lin et al., 2025) and Sa2VA (Yuan et al., 2025), we additionally train their open-sourced models with our Hour-RVOS train set. During training of our Memory-RVOS, we set 16 frames in one clip and train it in 60000 iterations with the AdamW optimizer (Loshchilov & Hutter, 2017) with a initial learning rate set at 5e-5.

## A.3 EXISTING BENCHMARK RESULTS

Table A1: Quantitative comparison compared to SOTA RVOS methods on Ref-YouTube-VOS (Seo et al., 2020), Ref-DAVIS17 (Khoreva et al., 2019) and MeViS (Ding et al., 2023) datasets.

| Method | Backbone | Ref-YouTube-VOS | | | Ref-DAVIS17 | | | MeViS | | |
|---|---|---|---|---|---|---|---|---|---|---|
| | | $\mathcal{J}\&\mathcal{F}$ | $\mathcal{J}$ | $\mathcal{F}$ | $\mathcal{J}\&\mathcal{F}$ | $\mathcal{J}$ | $\mathcal{F}$ | $\mathcal{J}\&\mathcal{F}$ | $\mathcal{J}$ | $\mathcal{F}$ |
| MTTR (Botach et al., 2022) | Video-Swin-T | 55.3 | 54.0 | 56.6 | - | - | - | 30.0 | 28.8 | 31.2 |
| ReferFormer (Wu et al., 2022) | Video-Swin-B | 62.9 | 61.3 | 64.6 | 61.1 | 58.1 | 64.1 | 31.0 | 29.8 | 32.2 |
| OnlineRefer (Wu et al., 2023) | Video-Swin-B | 62.9 | 61.0 | 64.7 | 62.4 | 59.1 | 65.6 | - | - | - |
| SgMg (Miao et al., 2023) | Video-Swin-B | 65.7 | 63.9 | 67.4 | 63.3 | 60.6 | 66.0 | - | - | - |
| HTML (Han et al., 2023) | Video-Swin-B | 63.4 | 61.5 | 65.2 | 62.1 | 59.2 | 65.1 | - | - | - |
| SOC (Luo et al., 2023) | Video-Swin-B | 66.0 | 64.1 | 67.9 | 64.2 | 61.0 | 67.4 | - | - | - |
| LoSh (Yuan et al., 2024) | Video-Swin-B | 67.2 | 65.4 | 69.0 | 64.3 | 61.8 | 66.8 | - | - | - |
| DsHMP (He & Ding, 2024) | Video-Swin-B | 67.1 | 65.0 | 69.1 | 64.9 | 61.7 | 68.1 | 46.4 | 43.0 | 49.8 |
| SSA (Pan et al., 2025) | CLIP | 64.3 | 62.2 | 66.4 | 67.3 | 64.0 | 70.7 | 48.6 | 44.0 | 53.2 |
| ReferDINO (Liang et al., 2025b) | GoundingDINO | 69.3 | 67.0 | 71.5 | 68.9 | 65.1 | 72.9 | 49.3 | 44.7 | 53.9 |
| Ours | Video-Swin-B | 67.5 | 65.2 | 69.8 | 65.8 | 69.5 | 62.1 | 47.0 | 44.4 | 49.6 |

To verify the effectiveness of our proposed Memory-RVOS method on existing RVOS datasets containing short-term videos, we conduct experiments on Ref-YouTube-VOS (Seo et al., 2020), Ref-DAVIS17 (Khoreva et al., 2019) and MeViS (Ding et al., 2023). As shown in Tab. A1, compared with RVOS methods using the same backbone (Video-Swin-B (Liu et al., 2022)), our Memory-RVOS achieves the best performance when evaluating on these datasets with short-term videos.

## A.4 FAILURE CASES

The failure cases when evaluating our proposed method on our Hour-RVOS dataset are shown in Fig. A1 (b). When it comes to extreme complex object interactions like "swapped with the cup on the right" in the failure case (a), our method cannot identify the correct cup after swapping cups. When objects undergo significant occlusions across long-temporal span in the failure case (b), our method cannot follow the unobstructed part of the vehicle. These failure cases sufficiently illustrate the tough challenges posed by our dataset including objects with dynamic changes over long-time, complex language expressions and their intricate correspondences. As shown in Tab. A2, we also list

Table A2: The definition of attributes counted in Fig. 3 (b).

| Attribute | Definition |
|---|---|
| BC | Background Clutter. The appearances of background and target object are similar. |
| DEF | Deformation. Target appearance deform complexly. |
| MB | Motion Blur. Boundaries of target object is blurred because of camera or object fast motion. |
| FM | Fast Motion. The per-frame motion of target is larger than 20 pixels, computed as the centroids Euclidean distance. |
| LR | Low Resolution. The average ratio between target box area and image area is smaller than 0.1. |
| OCC | Occlusion. The target is partially or fully occluded in the video. |
| OV | Out-of-view The target leaves the video frame completely. |
| SV | Scale Variation The ratio of any pair of bounding-box is outside of range [0.5,2.0]. |
| DB | Dynamic Background Background regions undergos deformation. |
| SC | Shape Complexity Boundaries of target object is complex. |
| AC | Appearance Change Significant appearance change, due to rotations and illumination changes. |
| LRA | Long-term Reappearance Target object reappears after disappearing for at least 100 frames. |
| CTC | Cross-temporal Confusion There are multiple different objects that are similar to targect object but do not appear at the same time. |

the definition of challenging attributes following (Hong et al., 2023) which are common in long-term videos and counted in Fig. 3.

## A.5 MORE ABLATION STUDIES

**Clip Division.** The key to processing any-length videos is to divide the entire videos into multiple clips. Therefore, we further explore the impact of clip size on the segmentation performance and efficiency. Specifically, we set the clip size to 4, 8, 16 due to the GPU memory limitation. As shown in Tab. A3, a larger clip size means more frames can be processed

Table A3: Ablation studies on the clip size.

| Clip size | FPS | Val (0-10 mins) | | | Val (over 10 mins) | | |
|---|---|---|---|---|---|---|---|
| | | $\mathcal{J}\&\mathcal{F}$ | $\mathcal{J}$ | $\mathcal{F}$ | $\mathcal{J}\&\mathcal{F}$ | $\mathcal{J}$ | $\mathcal{F}$ |
| 4 | 22 | 38.4 | 30.6 | 46.2 | 23.9 | 19.5 | 28.3 |
| 8 | 28 | 42.1 | 38.7 | 45.5 | 28.6 | 24.4 | 32.8 |
| 16 | 31 | 45.8 | 40.9 | 50.7 | 31.2 | 27.5 | 34.9 |

in parallel at one time and more informations can be associated. Therefore, as the clip size increases, the segmentation efficiency and accuracy also increase.

**Window Size.** We conduct experiments to explore the segmentation efficiency and accuracy under different window size settings. We set $W_l$ to $2, 4$, and $W_p$ to $4, 8$ respectively due to memory limitation as shown in Tab. A4. When $W_l$ is set to 2 and $W_p$ is set to $4, 8$, our method achieves higher efficiency but lower accuracy. When $W_l, W_p$ is set to $4, 8$, our method

Table A4: Ablation studies on the window size of long-term ($W_l$) and permanent ($W_p$) memory.

| $W_l$ | $W_p$ | FPS | Val (0-10 mins) | | | Val (over 10 mins) | | |
|---|---|---|---|---|---|---|---|---|
| | | | $\mathcal{J}\&\mathcal{F}$ | $\mathcal{J}$ | $\mathcal{F}$ | $\mathcal{J}\&\mathcal{F}$ | $\mathcal{J}$ | $\mathcal{F}$ |
| 2 | 4 | 36 | 41.1 | 36.8 | 45.4 | 26.7 | 23.2 | 30.2 |
| 2 | 8 | 35 | 43.5 | 39.4 | 47.6 | 29.4 | 26.7 | 32.1 |
| 4 | 8 | 31 | 45.8 | 40.9 | 50.7 | 31.2 | 27.5 | 34.9 |

achieves the balanced accuracy and efficiency. In the final test settings, we set the window size of long-term and permanent memory as $4$ and $8$ respectively.