# OpenReview forum: "A Needle In A Haystack: Referring Hour-Level Video Object Segmentation"
_ICLR.cc/2026/Conference — ICLR 2026 Conference Withdrawn Submission_

### Official Review · Reviewer_iizf · 2025-10-19

**Soundness:** 3
**Presentation:** 3
**Contribution:** 3
**Rating:** 6
**Confidence:** 5

**Summary:**

This paper addresses the limitation of existing Referring Video Object Segmentation (RVOS) datasets, which only cover short-term videos (5-60 seconds), by constructing the first Hour-level RVOS (Hour-RVOS) dataset. To tackle the unique challenges of Hour-RVOS, such as sparse object distribution and long-range correspondences, the authors propose a semi-online Hierarchical-Memory-Association method (Memory-RVOS), which includes a Hierarchical Memory Interleaved Propagation module and a Linguistic-Visual Dynamic Balance module. The method is designed to retain core target information and balance cross-modal tokens, enabling real-time segmentation of any-length videos. Comprehensive experiments compare Memory-RVOS with existing offline RVOS methods and Multimodal Large Language Models (MLLMs) on Hour-RVOS.

**Strengths:**

1. Reasonable idea and great motivation: The proposal of Hour-RVOS and Memory-RVOS directly targets real-world scenarios where long-duration videos (e.g., daily records, movies) require accurate referring segmentation, making the research both practical and innovative.
2. The authors conduct an in-depth analysis of Hour-RVOS, including comparisons with 6 existing RVOS datasets and fine-grained statistics, which fully present the dataset’s complexity and representativeness.
3. The proposed method, Memory-RVOS, is highly tailored to the challenges of Hour-RVOS. Ablation studies (Tables 3, 4) further confirm that each component contributes to performance improvements.
4. The experiments cover multiple dimensions: benchmarking against diverse baselines on Hour-RVOS; verifying generalization on existing short-term RVOS datasets; conducting comprehensive ablation studies.

**Weaknesses:**

1. The paper uses "A NEEDLE IN A HAYSTACK" to metaphorically describe the challenge of sparse object distribution in hour-level videos, but it lacks in-depth discussion. For example, it does not compare this challenge with similar "needle-in-haystack" problems in other video tasks.

   - Needle In A Video Haystack: A Scalable Synthetic Evaluator for Video MLLMs.
   -  Multimodal Needle in a Haystack: Benchmarking Long-Context Capability of Multimodal Large Language Models.
   - Re-thinking Temporal Search for Long-Form Video Understanding.

2. The paper introduces hierarchical memory, but it overlooks a significant body of prior work on hierarchical memory systems. For instance, earlier studies have investigated hierarchical memory in the domains of video segmentation.

   - XMem: Long-Term Video Object Segmentation with an Atkinson-Shiffrin Memory Model.

**Questions:**

1. Table A3 shows that when clip size increases from 4 to 16 (a 4x increase), FPS only rises from 22 to 31 (less than a 1.5x increase), but the paper does not explain this discrepancy.

---

### Official Review · Reviewer_h6TK · 2025-10-27

**Soundness:** 2
**Presentation:** 1
**Contribution:** 2
**Rating:** 4
**Confidence:** 4

**Summary:**

The paper introduces Hour-RVOS, the first referring video object segmentation dataset with any-length videos up to hour with richer language annotations (avg. 18.3 words) and multi-round interactions. The proposed dataset identifies two core challenges of long video RVOS: sparse object occurrence and long-range language-vision relationship. The paper further proposed a new semi-online method Memory-RVOS with hierarchical memory and a language-visual balance module. Experiments show that the proposed method outperforming current baselines on the new proposed Hour-RVOS dataset.

**Strengths:**

- The RVOS dataset proposed is substantially larger than existing datasets in average sequence length, total duration, number of masks, description complexity, and overall difficulty. It is also the first dataset to include multi-round semantic descriptions, which supports training and evaluating complex semantic segmentation in long videos.

- Evaluation on existing methods indicate the proposed dataset is sufficiently challenging.

- The overall logic of the paper looks good (though there are significant issues in its methodological clarity; see comments below).

**Weaknesses:**

- According to Tab. 1, the proposed Hour-RVOS contains similar / fewer objects and categories than some existing datasets. This may limit the generalization of models trained or evaluated on Hour-RVOS to open-world object scenarios. A deeper discussion of this limitation would strengthen the paper.

- The method description is unclear. Sections 3.2 and 3.3 present an imbalanced and sometimes illogical exposition of the pipeline. I recommend reorganizing these sections for clarity. Several design considerations are neither clearly justified in the method nor supported by targeted ablations. Please see “Questions” for my specific concerns.

- Many typos and figure issues, naming a few:

L137: “GoundingDINO” → “GroundingDINO”.

Fig. 1(b) caption: “Average expression length” → “Average duration”.

Fig. 4 lacks legends and annotations, making it difficult to match components to symbols in the text.

Figs. 4 and 5: symbol rendering errors (displayed as question marks).

Overall, figure notation and typography errors are frequent. A thorough proofreading pass is strongly recommended.

**Questions:**

**About the Hour-RVOS dataset**

- Per L343, the category distributions are consistent across train/val/test. In contrast, established VOS benchmarks (e.g., YouTube-VOS [1]) include both seen (overlapping with train) and unseen (not seen at all during training) categories in the test set to evaluate generalization abality to open-world objects. Given the proposed Hour-RVOS is not dramatically larger than YouTube-VOS w.r.t. object/category counts, could you consider a similar seen/unseen split (or an auxiliary evaluation) to assess more general generalization?

**About the proposed RVOS method**

- Section 3.2 (L241–L245) states that text features and object queries are fused and fed to MaskFormer; the resulting object tokens are then fused with memory tokens in the Hierarchical Memory Interleaved Propagation module, and the fused target object tokens are decoded into masks. However, Fig. 4 seems not consistent with this description. In Fig. 4, which components correspond to the object queries Q?, which one correspond to the “potential object tokens”?, and where are the “frame-mask level features”?

- Section 3.3 (L262): How exactly is the short-term memory updated? Please provide a precise description.

- Section 3.3 (L265): The “permanent memory” select tokens with the highest feature similarity from the long-term memory and also impose a window-size limit. This seems quite sparse—how do you ensure the selected tokens preserve sufficient object semantics?

- Section 3.3 (L282): The permanent tokens are compressed in the query encoding stage. What is the motivation for this compression, and how do you mitigate potential information loss?

- Section 3.3 (L295): “We place different memory tokens into different encoding process to adequately match visual-linguistic tokens in different temporal scale.” To my understanding, short-term tokens and the compressed permanent memory are fused with object queries during query encoding, while long-term memory tokens are fused later via cross-attention. What motivates this asymmetric design?

References
[1] Xu, Ning, et al. “YouTube-VOS: A large-scale video object segmentation benchmark.” arXiv:1809.03327 (2018).

---

### Official Review · Reviewer_Fj5M · 2025-10-27

**Soundness:** 2
**Presentation:** 2
**Contribution:** 2
**Rating:** 2
**Confidence:** 4

**Summary:**

This paper introduces an Hour-level RVOS dataset, which features sparse object distributions and long-range temporal correspondence across videos. To address the long-range object association challenge, the authors also propose a semi-online hierarchical memory association method (Memory-RVOS). The proposed approach achieves significantly better performance than existing RVOS methods on their constructed benchmark.

**Strengths:**

- The introduced Hour-RVOS dataset is the first benchmark designed specifically for hour-level RVOS tasks.
- The paper proposes a hierarchical memory association framework tailored for long-range object tracking and segmentation in RVOS. The method achieve much better performance than exisiting methods in the Hour-RVOS benchmark.

**Weaknesses:**

Dataset
- As a benchmark, the dataset size is relatively small (only 300 videos), compared with thousands in existing long-RVOS datasets (e.g., Long-RVCOS, MeViS, Ref-YouTube-VOS). Although the total number of frames is larger, the number and diversity of object instances remain limited. Moreover, it only contains 90 test videos.
- The duration statistics show that most videos are under 10 minutes, with only a small portion exceeding 30 minutes.
- The role of rich textual expressions is unclear. For RVOS benchmarks, it is debatable whether detailed multi-attribute language descriptions are necessary. In real-world use, expressions are often short and simple. Thus, metrics such as Average Words (Table 1) may not be meaningful. Similarly, the multi-round expressions could blur the boundary between visual reasoning and language-based re-identification. They may aid long-range association through text rather than through the RVOS model itself.

Method
- The description of Memory-RVOS is unclear and under-specified. For example, it is not specified which modules are trained, e.g., the linguistic encoder, Mask2Former, or other components? The loss functions are not detailed either.
- The hierarchical memory update and dynamic balance mechanisms are ambiguously described: are they applied during inference only, or during both training and inference?
- In Fig. 4(b), the design rationale for placing visual short-term and permanent memory in query encoding, while long-term memory and linguistic features are used as keys/values, is not well justified. Would switching permanent and long-term memory improve performance?

Experiments
- It is unclear whether the competing methods in Table 2 were fine-tuned on the Hour-RVOS dataset; if not, the comparison may be unfair.
- Ablation studies are conducted only on the validation set (30 videos) rather than the full test set.
- From Table 3, even the short-term memory variant outperforms previous SOTA methods, suggesting that gains may arise from dataset bias (e.g., rich semantics or multi-round expressions) or fine-tuning effects, rather than the proposed method itself.
- Although the authors claim real-time performance (Line 364), no inference time results are reported in the main paper.

**Questions:**

Stated in Weakness part.

**Details Of Ethics Concerns:**

The collected videos need ethics review.

---

### Official Review · Reviewer_kwao · 2025-10-30

**Soundness:** 2
**Presentation:** 3
**Contribution:** 2
**Rating:** 4
**Confidence:** 4

**Summary:**

The paper introduces Hour-RVOS, which targets second-to-hour videos (100.4h, 300 videos, 9,114 expressions). It highlights two challenges: Sparse object distribution and Long-range correspondence. To address this, the authors propose a semi-online method with hierarchical memory and language visual dynamic balance.

**Strengths:**

1. The dataset is well-designed, featuring videos of arbitrary length and multi-round semantic interactions, effectively capturing the complexity of real-world scenarios.
2. The proposed semi-online hierarchical memory association framework (Memory-RVOS) demonstrates clear effectiveness in modeling long-term cross-modal dependencies.
3. Experimental results provide strong evidence for the effectiveness of the proposed approach.

**Weaknesses:**

1. As shown in Figure 1(a), fewer than ten videos have a duration of one hour, and the average length is around 20 minutes. So, describing the dataset as hour-level appears to be somewhat of an overclaim.
2. Lines 313–314 state that the model uses “newly arrived linguistic features to highlight the visual tokens stored in memory”. However, this mechanism is not further elaborated — only the procedures for pruning noise tokens and reweighting linguistic tokens are discussed.
3. The semantic richness of the dataset is limited — the number of videos, expressions, and object categories does not offer a significant advantage over existing datasets.
4 .The paper lacks visualization results to demonstrate that the proposed modules effectively address the issues of sparse object distribution and long-range correspondence.

**Questions:**

1. What is the total number of words in Hour-RVOS and other RVOS dataset? Does the so-called rich-semantic expressions mainly refer to the average length of each expression?
2. How are the newly arrived linguistic features used to highlight the visual tokens stored in memory? Could the authors provide more detailed explanations?
3. Multi-round expressions are a key feature of Hour-RVOS. Could the authors provide statistics on how many referring expressions each video contains？
4. The Long-RVOS dataset also focuses on long-term referring video object segmentation. What are the main differences between Long-RVOS and Hour-RVOS?
5. When generating expressions using LLMs, what kinds of prompts were used? Were there fixed templates, and how were multiple aspects of an object combined into a single complete expression?

---

### Note · Authors · 2025-11-24

I have read and agree with the venue's withdrawal policy on behalf of myself and my co-authors.